# The Research Progress in Transforming Growth Factor-β2

**DOI:** 10.3390/cells12232739

**Published:** 2023-11-30

**Authors:** Meng-Yan Wang, Wen-Juan Liu, Le-Yi Wu, Gang Wang, Cheng-Lin Zhang, Jie Liu

**Affiliations:** 1Department of Pathophysiology, Shenzhen University Medical School, Shenzhen 518060, China; 2100243031@email.szu.edu.cn (M.-Y.W.); lwj0320@szu.edu.cn (W.-J.L.); 2200243053@email.szu.edu.cn (L.-Y.W.); liuj@szu.edu.cn (J.L.); 2Shenzhen Research Institute, The Chinese University of Hong Kong, Shenzhen 518000, China

**Keywords:** transforming growth factor-beta 2, regulation, signalling pathway, biological function

## Abstract

Transforming growth factor-beta 2 (TGF-β2), an important member of the TGF-β family, is a secreted protein that is involved in many biological processes, such as cell growth, proliferation, migration, and differentiation. TGF-β2 had been thought to be functionally identical to TGF-β1; however, an increasing number of recent studies uncovered the distinctive features of TGF-β2 in terms of its expression, activation, and biological functions. Mice deficient in TGF-β2 showed remarkable developmental abnormalities in multiple organs, especially the cardiovascular system. Dysregulation of TGF-β2 signalling was associated with tumorigenesis, eye diseases, cardiovascular diseases, immune disorders, as well as motor system diseases. Here, we provide a comprehensive review of the research progress in TGF-β2 to support further research on TGF-β2.

## 1. Introduction

The transforming growth factor-β (TGF-β) family is a superfamily of growth factors that play critical roles in regulating growth, development, immunity, inflammation, and tissue repair. To date, 33 members of the TGF-β superfamily have been identified in the human genome, including TGF-βs, bone morphogenetic proteins, growth differentiation factors, etc. [1]. The TGF-βs include three members, namely, TGF-β1, -β2, and -β3. Among them, the TGF-β1 homodimer is considered the prototype as it was the first TGF-β family protein to be biochemically characterised and readily available as a purified protein for experimental use [2,3]. The TGF-β signalling cascade is initiated by the binding of mature ligands to corresponsive receptors on the cell surface, forming a complex that activates Smad or non-Smad molecules to transduce signals into the nucleus and regulate the expression of target genes [4]. Due to their high similarity in protein sequence and structure, TGF-β1, -β2, and -β3 share similar receptors and intracellular signalling molecules. Most researchers tend to assume that the biological functions of TGF-β2 and TGF-β3 are equivalent to those of TGF-β1. However, a growing body of evidence has shown that TGF-β2 has many specificities that are different from those of TGF-β1 and TGF-β3. For example, TGF-β1 and TGF-β3 both have high binding affinities with the type 2 receptor (TβR2), whereas the affinity of TGF-β2 to TβR2 is much lower [5]. Further evidence from TGF-β1, TGF-β2, and TGF-β3 global knockout mice suggests that their physiological functions are significantly different (Table 1). About 60% of *Tgfb1^−/−^* mice die at the embryonic stage. Some postnatal *Tgfb1^−/−^* mice showed no obvious developmental abnormalities but eventually died around 20 days after birth due to a multifocal mixed inflammatory cell response, and organ failure [6]. *Tgfb3^−/−^* mice die within 20 h of birth with delayed lung development and defective palatogenesis [7]. In comparison, *Tgfb2^−/−^* mice exhibit more pronounced perinatal mortality and multiple developmental abnormalities that affect the cardiac, pulmonary, craniofacial, skeletal, palate, eye, inner ear, and urogenital organs [8]. The number of articles published each year on the topic of TGF-β2 has been increasing since 1987 when TGF-β2 was first described by Ikeda and colleagues (Figure 1A). Visualisation of the frequency of research keywords shows that the hot topics for TGF-β2 are its expression, activation, and roles in inflammation, tumorigenesis, differentiation, apoptosis, and extracellular matrix production in vivo and in vitro (Figure 1B). However, there is currently no available summary of the research progress on TGF-β2. Therefore, this review aims to provide an overview of recent advances in the expression, activation, regulation, and biological function of TGF-β2.

## 2. The Molecular Biology of TGF-β2

### 2.1. The Expression and Procession of TGF-β2

The amino acid sequence of TGF-β2 is highly conserved among species (Figure 2A). The human *TGFB2* gene is located on chromatin 1q41 and contains seven or eight exons. There are two isoforms of human *TGFB2* mRNA. Isoform 1 mRNA has a total length of 5952 bp and encodes a protein of 442 amino acids, while isoform 2 mRNA has a total length of 5868 bp and encodes a protein of 414 amino acids [10]. Isoform 1 has an additional 87 bp coding sequence that encodes 29 amino acids (tvcpvvttpsgsvgslcsrqsqvlcgyld) (https://www.ncbi.nlm.nih.gov/protein/NP_001129071.1, accessed on 1 October 2023). These amino acids replace an Asn (n) residue located at position 116 in the N-terminus of the TGF-β2 isoform 2 precursor protein (Figure 2B). Since the active form of TGF-β2 is located at the C-terminus, both isoforms express exactly the same mature TGF-β2. Similarly, mouse *Tgfb2* mRNA also has two isoforms, with isoform 2 containing a 29-amino-acid insertion (tvcpvvttpsgslgsfcsrqsqvlcgyld) replacing the Asn (n) at the 116 position of isoform 1. Similar to humans, both isoforms express the same mature TGF-β2. The additional 29 amino acids of human isoform 1 and mouse isoform 2 contain three cysteine residues, which are essential for the stable protein conformation, dimerisation, and activation of TGF-β2. However, it is unclear whether differences in the N-terminus of the two isoforms affect the shear and release of mature TGF-β2.

Similar to other TGF-β isoforms, the pre-propeptide of TGF-β2 consists of three fragments, namely, the N-terminal signal peptide, the latent-associated peptide (LAP), and the C-terminal mature fragment [4]. After the pre-propeptide leaves for the endoplasmic reticulum from the ribosome, the N-terminal signal peptide is cleaved, and the remaining fragment (propeptide) enters the lumen. The conjugation sites of LAP and the mature peptide can be cleaved by furin protease and fold to form a homodimeric or heterodimeric complex called the small latent complex (SLC). The processing of the SLC is accompanied by the cross-linking of the N-terminal LAP to other secreted proteins, including the latent TGF-β binding proteins (LTBPs) and trans-membrane leucine-rich repeat protein glycoprotein-A repetitions predominant protein (GARP) [11]. GARP appears to be restricted to regulatory T cells and platelets, whereas LTBPs are ubiquitously expressed, which suggests that LTBPs may have broader impacts on TGF-β functions [12,13,14]. The LTBP family consists of four isoforms (LTBP-1, -2, -3, and -4), among which LTBP-1 and -3 can efficiently bind to all three members of TGF-β. By contrast, LTBP-4 has a weak binding affinity with TGF-β1, whereas LTBP-2 cannot interact with any isoforms of TGF-β [15,16]. SLC and LTBP form a protein known as the large latent complex (LLC). The LLC is delivered to the Golgi apparatus after assembly in the endoplasmic reticulum. Then, LLC is exocytosed into secretory vesicles, where it is released into the extracellular matrix (ECM) [17]. The deposition of latent TGF-β on the ECM is mediated by LTBP, which provides the scaffolding that binds latent TGF-β to ECM proteins such as fibrillins and fibronectins (Figure 3).

### 2.2. The Activation of TGF-β2

In the LLC, the LAP covers the key amino acids of the C-terminal dimer that are used to interact with its receptors, and this covering can cause the inactivation of the mature dimer ligand [17]. Therefore, TGF-β2 activation requires the dissociation or proteolysis of LLC in the ECM to release the dimeric TGF-β2 ligand. A common feature of these activators is that they all indicate perturbation of the ECM. Here, we discuss some established TGF-β2 activators (Table 2).

*Integrin*: The LAPs of TGF-β1 and -β3 contain the integrin-binding sequence RGD (Arg-Gly-Asp). Mice lacking the integrins αvβ6 and αvβ8 have phenotypic defects similar to those of TGF-β1 and -β3 knockout mice. However, TGF-β2 cannot be activated by integrin because TGF-β2 does not contain the RGD sequence in the LAP [18,19].

*Proteases*: Many proteases can activate TGF-β, including plasmin, serine protease, matrix metalloproteinase (MMP-2, MMP-9), plasma kallikrein, etc. [20,21,22,23]. These proteases activate LLC by directly digesting the LAP or indirectly digesting ECM components. Prostate-specific antigen (PSA) is a serine protease. It is generally considered to be a clinical marker of prostate cancer. In fact, PSA could play a role in the activation of latent TGF-β. In contrast to plasmin, which is known to activate both latent TGF-β1 and TGF-β2, PSA was found to activate only latent TGF-β2 in prostate epithelial cells [24].

*Thrombospondin-1*: Thrombospondin-1 (TSP-1) is a component of the ECM. Physiologically, TSP-1 is secreted by platelets, but a number of other cells are also capable of producing it, including smooth muscle cells, astrocytes, endothelial cells, and various tumour cells [25]. TSP-1 has a binding site for LAP and serves as an activator of TGF-β. In antigen-presenting cells and glioma cells, TSP-1 has been reported to efficiently activate TGF-β2 [26,27].

*Reactive oxygen species*: When latent TGF-β was exposed to Fe(III)/ascorbate-generated reactive oxygen species (ROS), only latent TGF-β1 showed significant activation, while latent TGF-β2 and latent TGF-β3 did not. This is because the direct target of ROS is methionine 253, a methionine unique to LAP-β1, where its side chain is oxidised, leading to conformational alterations that result in the activation of latent TGF-β1 [28].

*pH:* Treating NRK-49F and AKR-MCA cell-conditioned medium with extreme pH (1.5 or 12) resulted in significant activation of TGF-β1, whereas mild acidity (pH 4.5) resulted in only 20–30% of the effect achieved with pH 1.5, and little or no TGF-β1 activity was produced at a pH between 5.5 and 7.5 [29]. Another study showed that the sensitivity of recombinant human TGF-β1 and TGF-β2 to acids and bases is basically the same. Both isoforms were activated in the acidic (pH 3.1–4.1) and alkaline (pH 11.0–11.9) ranges. In contrast, recombinant chicken TGF-β3 was activated under acidic (pH 2.5–3.1) and alkaline (pH 10.0–12.3) conditions [30].

*Heat*: Thermal activation of latent TGF-β was measured with the growth inhibition of the mink lung cell line CCL-64. Complete activation of recombinant latent TGF-β1 was observed at 70 °C for 10 min, 75 °C for 5 min, and 85–90 °C for 1 min. The thermal stability of the different latent forms was compared at 25–100 °C for 5 min. Recombinant latent TGF-β1, TGF-β2, and TGF-β3 showed similar activation trends at 25–75 °C, but latent TGF-β2 and TGF-β3 showed a sustained increase in activation from 80 to 100 °C, whereas the activation of latent TGF-β1 decreased [30]. However, the impact of heat on TGF-βs activity has not been verified with in vivo experiments, and their physiological and pathological significance in the body needs to be further investigated.

*Mechanical force*: In an early report, researchers found that shear stress-induced an overall decrease in TGF-β pathway molecules, including TGF-β2, TGF-β3, TβR1, and TβR2 expression in tendon cells [31]. However, TGF-β1 was upregulated in response to stirring or shear force. Subsequently, TGF-β1 was also shown to be strongly activated when subjected to mechanical forces in vitro and in vivo. One possible molecular mechanism by which stirring and shear may influence the activation of latent TGF-β1 is thiol-disulphide exchange [32]. The reason for the difference in the mechanical force required to activate TGF-β1, TGF-β2, and TGF-β3 is still unclear. Whether this difference is caused by the difference in some amino acid sites of LAP requires further experimental evidence.

**Table 2 cells-12-02739-t002:** Comparison of factors affecting the activation of three isoforms of TGF-β.

Activators	TGF-β1	TGF-β2	TGF-β3	Reference
Integrins	Activate	Unable to activate	Activate	[18,19]
Proteases	Activate	Activate	Activate	[24]
TSP-1	Activate	Activate	NA	[26,27]
ROS	Activate	Unable to activate	Unable to activate	[28]
pH				
pH 3.1–4.1 pH 11.0–11.9	Activate	Activate	NA	[30]
pH 2.5–3.1	NA	NA	Activate	
pH 10.0–12.3				
Heat				
70 °C (10 min)	Fully activate	NA	NA	[30]
75 °C (5 min)	Fully activate	Partially activate	Partially activate	
85–90 °C (1 min)	Fully activate	NA	NA	
100 °C (5 min)	Partially inactive	Fully activate	Fully activate	
Mechanical force	Activate	NA	NA	[31,32]

NA, not available.

### 2.3. The Signalling Pathway of TGF-β2

#### Receptors

There are three types of TGF-βs receptors, namely, TβR1, TβR2, and TβR3 (betaglycan). TβR1 and TβR2 were found to have serine/threonine and tyrosine kinase activity, while TβR3 showed no kinase activity. TGF-β1-3 has different affinities for the three receptors, so the way they activate downstream signalling pathways is also different [33]. TGF-β1 and TGF-β3 have strong affinity with TβR2. When TβR1 and TβR2 coexist, TGF-β1 and TGF-β3 first bind to TβR2, induce the autophosphorylation of TβR2, and then recruit TβR1 to form a tetrameric complex. Phosphorylation of the glycine–serine enrichment domain (GS-domain) of TβR1 leads to the activation of the downstream signalling pathway. Compared with TGF-β1 and TGF-β3, the affinity of TGF-β2 for TβR2 is much lower [5]. The reason for the different affinity between TGF-β1-3 and the receptors may be that TGF-β1 and TGF-β3 contain Arg25, Val92, and Arg94, while these three amino acids in TGF-β2 are replaced by Lys25, Ile92, and Lys94 [34]. TβR3 is a necessary co-receptor for TGF-β2 to perform some physiological regulatory functions, and its high affinity with TGF-β2 enhances the response of target cells to TGF-β2 [35,36]. After interacting with active TGF-β2, TβR3 can recruit and combine with TβR2, which can further recruit TβR1 to form a complex. Thereafter, TβR3 dissociates from the above complex and eventually forms the TGF-β2/TβR1/TβR2 complex to carry out the activation of the downstream signalling pathway [37]. In vivo studies have shown phenotypic similarities between *Tgfb2^−/−^* and *Tgfbr3^−/−^* mice, such as embryonic lethality and cardiac dysplasia [38,39]. In vitro studies have also shown that mouse embryonic fibroblasts with TβR3 knockdown are significantly less sensitive to the TGF-β2 response, while this effect is not observed for other TGF-β ligands [38]. These studies suggest that the cardiac developmental defects caused by TβR3 deletion mutations may be related to the weakened signalling of TGF-β2, which further indicates that TβR3 is an important co-receptor for TGF-β2 to play the regulatory function of cardiac development. However, it is important to note that TβR3 does not always have a positive effect on TGF-β2 signalling. TβR3 has a soluble form, which is the cleaved extracellular fragments of full-length TβR3. Researchers found that adding soluble TβR3 to mink lung epithelial cells strongly inhibited TGF-β2 signal transduction. The addition of 0.6 nM soluble TβR3 was able to reduce the binding of TGF-β2 to TβR1 and TβR2 by 90% and to reduce the growth inhibitory effect of 5 pM TGF-β2 by 50% [40]. Further experiments are needed to determine whether TβR3 can play a dual regulatory function in vivo and to prove its utility as a TGF-β2 modulator. In addition, not all TGF-β2 regulatory processes depend on TβR3. In the absence of TβR3, the TGF-β2 signal could also be partially transduced by TβR2-B in a Smad-dependent manner. TβR2-B is an alternatively spliced variant of TβR2 and is expressed in cells that are sensitive to TGF-β2 such as osteoblasts and mesenchymal precursor cells. Compared with TGF-β1, only minor responsiveness to TGF-β2 was observed in L6 skeletal muscle cells that did not express either TβR3 or TβR2-B. However, transfection of TβR3 or TβR2-B significantly increased the response of L6 cells to TGF-β2 (Figure 4) [41].

### 2.4. The Regulation of TGF-β2 Expression (Table 3)

#### 2.4.1. Transcription Factors

*Cyclic AMP-responsive element-binding protein H*: Cyclic AMP responsive element binding protein H (CREBH) is a transcription factor that is enriched in the liver and is known to regulate the hepatic acute phase response and lipid homeostasis. Patients with hepatitis C had higher serum TGF-β2 than a healthy population, and the concentration of TGF-β2 was positively correlated with hepatic fibrosis stages. Meanwhile, the mRNA level of *Tgfb2* was increased after hepatic stellate cells were infected with the hepatitis C virus. Silencing CREBH inhibits, while overexpressing CREBH upregulates, the transcription of TGF-β2. Using chromatin immunoprecipitation assays, one study further identified a CREBH binding element at −49 to −43 in the *TGFB2* promoter region [42].

*Oestrogen-related receptor*: Oestrogen-related receptor γ (ERRγ) is a nuclear receptor that serves as a constitutive activator of transcription. In CCl_4_-induced acute liver injury models, TGF-β2 expression and secretion were increased in an ERRγ-dependent manner. Promoter assays further demonstrated that the promoter region of the human *TGFB2* gene contains an ERRγ binding element at −1686 to −1676, which mediates the direct transcriptional regulation of *TGFB2* by ERRγ [43].

*Homeobox*: Homeobox B7 (HoxB7) is a transcription factor that is critically involved in tumorigenesis. TGF-β2 expression was upregulated in four MMTV-HoxB7/Her2 transgenic mouse tumour cell lines and two breast cancer cell lines after HOXB7 overexpression, whereas TGF-β2 expression was reduced in HoxB7-depleted cells. Using luciferase and chromatin immunoprecipitation assays, one study further identified that HoxB7 can directly bind to the promoter region of TGF-β2 and activate its transcription [44]. HoxA10 is another member of the homeobox superfamily involved in tumour progression. TGF-β2 is a target gene of HoxA10. Overexpression of HoxA10 increases TGF-β2 transcription by interacting with tandem cis-elements in the promoter [45].

*Snail*: Snail is a pro-fibrotic transcription factor implicated in epithelial-to-mesenchymal transition (EMT) and tumour metastasis. In pancreatic cancer-susceptible mice (Kras mutation), pancreatic acinar cell-specific Snail overexpression upregulated the expression of TGF-β2 and enhanced pancreatic fibrosis, while the expression of TGF-β1 and TGF-β3 were not significantly changed. Similarly, the overexpression of Snail in AsPC1 and Panc1 cell lines selectively upregulates TGF-β2 with no effect on TGF-β1 or TGF-β3 [46].

*Activating transcription factor*: Activating transcription factor 3 (ATF3) is a member of the activating transcription factor/cAMP-responsive element-binding protein family of transcription factors. In HUVECs, ATF3 selectively binds to the promoter region of *TGFB2*, but not *TGFB1*, thereby activating TGF-β2 expression and facilitating endothelial to mesenchymal transformation [47]. In addition, the direct binding of phospho-ATF2 to the *TGFB2* promoter region is one of the mechanisms by which all-trans-retinoic acid increases TGF-β2 expression in intestinal epithelial cells [48].

*Peroxisome proliferator-activated receptors*: Peroxisome proliferator-activated receptors (PPARs) are a group of transcription factors that play essential roles in cellular differentiation, development, and metabolism. The PPARγ agonist efatutazone negatively regulated the mRNA and protein expression of TGF-β2 in non-small cell lung cancer cell lines [49]. In contrast, fenofibrate, a potent PPARα agonist, significantly increased TGF-β2 secretion in mouse subcutaneous white adipose tissue cells and partially explained the mechanism for exercise-induced lactate regulation of TGF-β2 [50].

*Regulatory factor X*: Regulatory factor X (RFX) is another transcription factor that plays an inhibitory role in *TGFB2* transcription. Researchers found that *TGFB2* promoter activity was decreased in the neuroblastoma cell line overexpressing RFX. Their study further demonstrated that the inhibitory effect of RFX on *TGFB2* transcription is mediated by its direct binding to the promoter region (−113 to −100) of the human *TGFB2* gene (Table 3) [51].

#### 2.4.2. Noncoding RNA

*MicroRNAs*: MicroRNAs (miRNAs) are a class of endogenous noncoding RNAs with ~22 nt that can bind with complementary sequences on mRNAs, resulting in mRNA degradation and/or translational inhibition [52]. TGF-β2 expression is regulated by a variety of miRNAs. For example, *TGFB2* is a direct target of miR-7-5p with the RNA pull-down assay and the dual luciferase reporter gene assay. In an acidic tumour microenvironment, the expression of miR-7-5p was reduced, alleviating its suppressive effect on the *TGFB2* mRNA 3′ untranslated region (3′UTR). As a result, the metastatic potential of lung cancer increased [53]. miR-148a downregulated the expression of *TGFB2* in gastric cancer cells, which contributed to the inhibition of tumour cell proliferation and metastasis [54]. miR-193a-3p is another miRNA that can suppress *TGFB2* expression by binding directly to the 3′UTR of *TGFB2*. miR-193a-3p expression was significantly reduced along with upregulated TGF-β2 levels in human heart tissue from congenital heart disease. miR-193a-3p mimics suppressed TGF-β2 expression in H9C2 cells, while miR-193a-3p inhibitor increased TGF-β2 expression [55]. miR-29b-3p and miR-29c-3p can target to the 3′UTR region of *Tgfb2* in mouse cardiac fibroblasts [56]. Similarly, miR-200a binds to the 3′UTR of *Tgfb2* and suppresses its expression in rat proximal tubular epithelial cells to prevent renal fibrosis [57]. miR-148b downregulates TGF-β2 to promote angiogenesis and inhibit endothelial-to-mesenchymal transition during skin wound healing [58]. *Tgfb2* mRNA expression was significantly downregulated in miR-31 mimic-treated mouse keratinocyte cells. In “Hairpoor” mice, miR-31 downregulation was accompanied by significant upregulation of TGF-β2, which can cause abnormal hair cycle in hair loss mutant mice. In contrast, there was no change in *Tgfb1* mRNA expression. In that study, the researchers further identified that the 3′ UTR at 4161~4181 bp of *Tgfb2* is the direct target of miR-31 in keratinocytes [59]. In addition, *Tgfb2* inhibition by miR-193b resulted in early chondrogenesis defection in chondrogenic ATDC5 cells [60]. miR-466a-3p targets *Tgfb2* in mouse CD4+T cells to inhibit its effect in inducing regulatory T cell differentiation from naive CD4 cells [61].

*Circular RNAs*: Circular RNAs (circRNAs) are a type of noncoding RNAs that play crucial roles in various pathological processes. circUbe2k can promote TGF-β2 expression by sponge miR-149-5p in the development of hepatic fibrosis [62]. circ_0001293 can clear the upstream inhibitory regulator (miR-8114) of TGF-β2 and increase its expression level. This circ_0001293/miR-8114/TGF-β2 axis has an anti-inflammatory effect in astrocytes, which might be a potential therapeutic target for epilepsy [63]. In addition, circRIP2 can sponge miR-1305 to upregulate TGF-β2, thereby inducing EMT and promoting proliferation and metastasis of bladder cancer cells [64].

*Long noncoding RNA*: Similar to the regulatory mechanism of circRNAs, there is another noncoding RNA, named long noncoding RNA (lncRNA). Increased expression of lncRNA cancer susceptibility candidate 9 (lncRNA CASC9) is associated with tumorigenesis and cancer progression. lncRNA CASC9 could upregulate the expression of TGF-β2 by sponging miR-758-3p, thereby accelerating bladder cancer progression [65]. In addition, another recent study reported that lncMIAT promotes markers of liver fibrosis with the upregulation of TGF-β2 by competitive sponging of miR-16-5p [66]. From the studies above, it is not difficult to speculate that the inhibition of TGF-β2 expression from the perspective of miRNA could be a good strategy for the treatment of a variety of diseases.

#### 2.4.3. External Stimulus

*Glycoproteins*: Glypican-3 (GPC3) is a cell surface glycoprotein that is highly expressed in hepatocellular carcinoma cells. According to Sun and co-workers, GPC3 silencing is correlated significantly with increased TGF-β2 expression, while having no significant effect on TGF-β1 and TGF-β3 expression. The antiproliferative effects of GPC3 silencing were partially attenuated by co-transfection of siRNA-GPC3 with siRNA-TGF-β2, indicating that TGF-β2 is a critical target of GPC3 in inducing cell proliferation [67]. Another study suggested that glycated collagen in the cardiac interstitium strongly increases the expression of TGF-β2, which is important for the formation of myofibroblasts and the fibrosis that occurs in diabetic cardiomyopathy [68]. However, the mechanism of the above two glycoproteins on TGF-β2 expression has not been clearly elucidated.

*Lipoprotein*: High-density lipoproteins (HDLs) are known as cholesterol carriers that have cardiovascular protective properties. Using cDNA array analysis, Norata et al. identified that TGF-β2, but not TGF-β1 or TGF-β3, is selectively increased by HDL in endothelial cells in a PI3K/Akt-dependent manner [69].

*Cytokines*: IL-4 and IL-13 stimulated the release of TGF-β2, but not TGF-β1 or TGF-β3, from human bronchial epithelial cells in a time- and concentration-dependent manner. IFN-γ was found to antagonise the stimulatory effects of IL-4 and IL-13 on TGF-β2 expression and secretion [70,71]. In addition, there was a trend towards a decrease in TGF-β2 mRNA in both *Il1b^−/−^* and *Il1r1^−/−^* mice, suggesting that IL-1β may have a positive role in TGF-β2 expression [72].

*H_2_O_2_*: There was a significant increase in TGF-β2 protein levels at the molecular weight of 12.5 kD after the treatment of human umbilical vein endothelial cells (HUVECs) with 10 μM H_2_O_2_ for 72 h. H_2_O_2_-induced phosphorylation of p38 MAPK might be a potential mechanism for the increase in TGF-β2 expression [73].

*Hypoxia*: Hypoxia (1%O_2_) can also increase the expression of TGF-β2 at both mRNA and protein levels in HUVECs. Mechanistically, Smad binding elements (CAGA box) at the −77 to −40 bp region of the *TGFB2* promoter are activated under hypoxia, which can enhance its transcription [74].

*Lactate*: TGF-β2 expression is closely linked to lactate metabolism. Tumour cells highly express lactate dehydrogenase type A (LDH-A), a key metabolic enzyme that catalyses the conversion of pyruvate into lactate. Small interfering RNA against LDH-A (siLDH-A) concentration-dependently suppressed TGF-β2 expression in HTZ-349 glioma cells, whereas lactate induced it [75]. Another study further revealed that lactate promotes TGF-β2 expression by inducing TSP-1, a TGF-β2 activating protein. The authors further found that both TSP-1 and TGF-β2 levels were reduced in glioma cells treated with siLDH-A and that the inhibition of TSP-1 results in a reduction in TGF-β2. Conversely, reduced levels of TGF-β2 protein can be rescued with the addition of synthetic TSP-1 [27].

*Mechanical forces*: Several kinds of mechanical forces can also affect TGF-β2 expression, including cell density and shear stress. A study on the effect of cell density on TGF-β2 in murine corneal epithelial cells found that TGF-β2 was only detected in conditioned media from cells seeded at 50,000 cells/cm^2^, but not at 500 or 5000 cells/cm^2^ [76]. When comparing sheared and unsheared tendon cells, unidirectional shear stress can downregulate the expression of TGF-β2 using cDNA microarrays and Northern blot analyses [31]. However, the mechanism of these mechanical factors on TGF-β2 is not yet clear.

**Table 3 cells-12-02739-t003:** The regulation of TGF-β2 expression.

	Effect	Mechanism	Diseases or Cell Types	Reference
External stimulus				
Glypican 3	Downregulate	NA	Hepatocellular carcinoma	[67]
Glycated- collagen	Upregulate	NA	Diabetic- cardiomyopathy	[68]
HDL	Upregulate	Activate PI3K/Akt	Atherosclerosis	[69]
TGF-β1	Downregulate	NA	Fibroblastic cell	[77]
TGF-β2	Upregulate	Inhibit Smad7	Necrotising enterocolitis	[78]
IL-4 IL-13	Upregulate	NA	Asthma	[70]
IFN-γ	Downregulate	Activate JAK-STAT	Retinal diseases	[71]
IL-1β	Upregulate	NA	Renal fibrosis	[72]
H_2_O_2_	Upregulate	Activate p38 MAPK	Systemic inflammation	[73]
Hypoxia	Upregulate	Activate CAGA box	Endothelial cell	[74]
Lactate	Upregulate	Activate by thrombospondin-1	Glioma	[27]
Cell density	Upregulate	NA	Corneal epithelial	[76]
Shear stress	Downregulate	NA	Tendon cell	[31]
Transcription factors				
CREBH	Upregulate	Bind promoter region at −49 to −43	Hepatitis C	[42]
ERRγ	Upregulate	Bind promoter region at −1686 to −1676	Acute liver injury	[43]
HOXB7	Upregulate	Binding region unknown	Breast cancer	[44]
HOXA10	Upregulate	Binding region unknown	Acute myeloid leukemia	[45]
Snail	Upregulate	Binding region unknown	Pancreatic cancer	[46]
ATF3	Upregulate	Binding region unknown	Vascular diseases	[47]
ATF2	Upregulate	Binding region unknown	Intestinal epithelial cells	[48]
PPARα	Upregulate	Binding region unknown	Glycolipid metabolism	[50]
PPARγ	Downregulate	Binding region unknown	Nonsmall cell lung cancer	[49]
RFX	Downregulate	Bind promoter region at −113 to −100	Neuroblastoma	[51]
Noncoding RNA				
miR-7-5p	Downregulate	Bind 3′UTR	Lung cancer metastasis	[53]
miR-148a	Downregulate	Bind 3′UTR	Gastric cancer	[54]
miR-193a-3p	Downregulate	Bind 3′UTR	Congenital heart disease	[55]
miR-29b/ 29c-3p	Downregulate	Bind 3′UTR	Fibroblast	[56]
miR-200a	Downregulate	Bind 3′UTR	Renal fibrogenesis	[57]
miR-148b	Downregulate	Bind 3′UTR	Skin wound healing	[58]
miR-31	Downregulate	Bind 3′UTR	Hairpoor	[59]
miR-193b	Downregulate	Bind 3′UTR	Chondrogenesis	[60]
miR-466a	Downregulate	Bind 3′UTR	Allogeneic transplantation	[61]
circUbe2k	Upregulate	Sponge miR-149-5p	Hepatic fibrosis	[62]
circ_0001293	Upregulate	Sponge miR-8114	Epilepsy	[63]
circRIP2	Upregulate	Sponge miR-1305	Bladder cancer	[64]
lncRNA CASC9	Upregulate	Sponge miR-758-3p	Bladder cancer	[65]
LncMIAT	Upregulate	Sponge miR-16-5p	Liver fibrosis	[66]

NA, not available.

## 3. The Physiological and Pathological Roles of TGF-β2

As TGF-β2 and its receptors are ubiquitously expressed in the body, the physiological and pathological roles of TGF-β2 are diverse. Apart from embryonic development, TGF-β2 is involved in the pathogenesis of diseases of the eye system, cardiovascular system, motor system, and immune system, as well as tumorigenesis (Table 4).

### 3.1. Eye System

Of the three TGF-βs, TGF-β2 is the predominant isoform in the neural retina and retinal pigment epithelium [104]. TGF-β2-induced EMT of retinal pigment epithelium cells is well-established as a major mechanism responsible for the pathogenesis of proliferative retinal diseases [79]. Nintedanib, a tyrosine kinase inhibitor that has anti-inflammation and anti-fibrosis effects, prevents TGF-β2-driven EMT in retinal pigment epithelium cells and may serve as a potential drug for the treatment of proliferative vitreoretinopathies [80]. Similarly, alpha-Klotho, an anti-ageing protein that diminishes with age, reversed EMT and senescence-like morphological alterations triggered by TGF-β2 in retinal pigment epithelium cells [81]. In addition, TGF-β2 has also been shown to be involved in age-related cataracts by promoting EMT and cell proliferation and inhibiting apoptosis in lens epithelial cells (LECs) [82]. Posterior capsular opacification, also known as fibrotic cataract, is partially caused by pathological EMT of LECs. During the EMT progression of LECs, TGF-β2 promoted autophagic flux in a time-dependent manner, which has been reported to play an important role in fibrotic disorders. Rapamycin-induced autophagic activation enhanced the TGF-β2-induced fibrotic response in LECs, whereas pharmacological inhibition of autophagy blunted TGF-β2-induced EMT of LECs, suggesting that increased autophagy is partially responsible for the pro-EMT effect of TGF-β2 in LECs [83]. Interestingly, TGF-β2 increased the expression of the tumour protein p53-inducible nuclear protein2, the knocking down of which blocked TGF-β2-induced autophagy and EMT, suggesting that tumour protein p53-inducible nuclear protein2 is a downstream regulator of autophagy flux induced by TGF-β2 and represents a novel target for the treatment of posterior capsular opacification [84].

### 3.2. Cardiovascular System

TGF-β2 plays a more prominent role in heart development. For example, Bartram et al. examined the heart of *Tgfb2^−/−^* mouse embryos from 11.5 to 18.5 days and detected obvious abnormalities in the outflow tract and atrioventricular channel. These phenotypes reflect the defects in looping, myocardialisation, differentiation of the endocardial cushions, and apoptosis [39]. Azhar et al. also found that the differentiation of endocardial cushions is mainly regulated by TGF-β2. They cultured atrioventricular canal explants from *Tgfb1^−/−^*, *Tgfb2^−/−^*, and *Tgfb3^−/−^* mice on collagen gels and found that TGF-β2 is the only ligand whose absence affects the EMT of the cardiac cushions [105]. The important role of TGF-β2 in vascular development was also highlighted. Researchers used double mutant mice with mutants of fibrillin 1 and TGF-β1, -β2, or -β3 and found that the deletion of TGF-β2, but not TGF-β1 or TGF-β3, resulted in 80% of the offspring dying. Mouse death was associated with hyperplasia of the aortic valve leaflets, aortic regurgitation, and enlargement of the aortic root, etc. [106].

In addition to its role in cardiovascular development, TGF-β2 has also been implicated in the pathological remodelling of the cardiovascular system. In heart diseases, TGF-β2 may be involved in promoting myocardial fibrosis. For example, TGF-β2 acted as a target gene of miR-29b-3p and miR-29c-3p in mouse cardiac fibroblasts, inhibiting the expression of genes associated with fibrosis such as COL1A1, COL3A1, and α-SMA [56]. Lysyl oxidation-like 2 (Loxl2), which is upregulated in the cardiac interstitium and is associated with collagen cross-linking and cardiac dysfunction, promotes the production of TGF-β2 in cardiac fibroblasts via the PI3K/AKT pathway. TGF-β2 is responsible for the transformation of fibroblasts into myofibroblasts induced by Loxl2 [85]. In vascular diseases, aortic aneurysm is a life-threatening feature of fibulin-4 mutated mice that could be reversed with the administration of a TGF-β-neutralising antibody. Further evidence suggests that TGF-β1 upregulation is marginal, whereas TGF-β2 upregulation is more obvious in aortic smooth muscle cells and the plasma of fibulin-4-deficient mice. More importantly, TGF-β2 levels were reduced after treatment with losartan, which is a known anti-aneurysm drug. These results suggest a potential role of TGF-β2 in the pathogenesis of aortic aneurysms [86]. In human carotid plaques, TGF-β2, but not TGF-β1 or TGF-β3, was the most abundant isoform. TGF-β2 expression in asymptomatic plaques was inversely correlated with matrix degradation and inflammation. The risk of cardiovascular events was lower in patients with plaques with high levels of TGF-β2. In vitro, TGF-β2 pretreatment reduced MMPs and monocyte chemoattractant protein-1, which may partially contribute to its protective role in the stabilisation of plaques [87]. Taken together, the above studies suggest that TGF-β2, rather than TGF-β1 or TGF-β3, may be the major player in the cardiovascular system.

### 3.3. Motor System

TGF-β2 is associated with the development or diseases of the motor system, which includes cartilage, tendon, bone, and skeletal muscle. In the articular cartilage isolated from neonatal mice, TGF-β2 was found to increase cell–cell communication between chondrocytes [88]. In chondrogenic ATDC5 cells of mice, miR-193b was upregulated and inhibited early chondrogenesis. The researchers further identified *Tgfb2* and *Tgfbr3* as the major target genes of miR-193b in the chondrogenesis of ATDC5 cells. The anti-chondrogenic effect of the miR-193b mimic was reversed with TGF-β2 treatment in a dose-dependent manner, indicating a crucial role of TGF-β2 in early chondrogenesis [60]. In human chondrocytes, TGF-β2 is involved in preserving the chondrocyte phenotype under hypoxic conditions in vitro [89]. Interestingly, in cultured human osteoarthritic cartilage, TGF-β2 was also found to suppress the expression of MMP and differentiation-related genes, thereby suppressing collagen resorption and chondrocyte differentiation [90]. These studies suggest a positive role of TGF-β2 in early chondrogenesis and maintenance of the chondrocyte phenotype. However, in contrast to its effects on chondrocytes and cartilage, TGF-β2 negatively affects skeletal muscle cells and bone formation. For example, in chicken skeletal muscle satellite cells, *Tgfb2*, which was identified as a target gene of miR-200a-3p, suppressed differentiation and proliferation and accelerated apoptosis of skeletal muscle satellite cells [91]. The osteogenic differentiation of human dental follicle stem cells was suppressed under inflammatory conditions with an obvious upregulation of TGF-β2. The inhibition of TGF-β2 enabled dental follicle stem cells to undergo osteogenic differentiation following lipopolysaccharide stimulation [92]. Tendons are important parts of the musculoskeletal system that are responsible for transferring forces from muscle to bone. miR-378a, by targeting *Tgfb2*, inhibits collagen and extracellular matrix production, thereby exerting an inhibitory effect on tendon repair both in vitro and in vivo. These results suggest that TGF-β2 is beneficial for tendon repair [93]. Another study also confirmed that treatment with TGF-β2 resulted in a significant upregulation of the expression of collagens, extracellular matrix molecules, and growth factors. This suggests that TGF-β2 is involved in promoting tendon healing by increasing the production of paracrine factors and extracellular matrix molecules [94]. Taken together, these studies suggest that TGF-β2 plays a positive regulatory role in chondrocyte activity and tendon repair and a negative regulatory role in skeletal muscle cell activity and bone formation.

### 3.4. Immune System

TGF-β is involved in maintaining the homeostasis of the immune system by regulating the activation of immune cells and the secretion of cytokines. Increasing evidence suggests that TGF-β1 plays an important role in the maintenance of immune homeostasis; however, the immunological role of TGF-β2 is not yet fully understood. Molecular evolutionary analyses revealed that lamprey contain two members of TGF-β, i.e., TGF-β2 and TGF-β3, but not TGF-β1, representing ancestors in vertebrates. Then, transcriptional expression patterns further showed that lamprey TGF-β2 is upregulated more rapidly and significantly than TGF-β3 during lipopolysaccharide stimulation. These results suggest that lamprey TGF-β2 may play a more important role in the innate immune response in lamprey [95]. In rodents, TGF-β2, but not TGF-β1, is expressed in neuron-glial antigen 2 glia, which are necessary for maintaining immune homeostasis in the brain. TGF-β2 significantly upregulated two microglial checkpoint genes, i.e., CX3C chemokine receptor 1 (CX3CR1) and Tmem119, and reduced pro-inflammatory mediators both in vitro and in vivo. That study demonstrated that glial-derived TGF-β2 is an important modulator of the CX3CR1-regulated innate immune response in the brain [96]. TGF-β2 can also be secreted by neural precursor cells and direct the transcriptional reprogramming of dendritic cells. Intrathecal administration of TGF-β2 resulted in the amelioration of autoimmune encephalitis, while transplantation of *Tgfb2^−/−^* neural precursor cells into mice with experimental autoimmune encephalomyelitis had no effect on immune response cell accumulation [97]. In addition, the presence of TGF-β2 in breast milk is thought to be associated with the maturation of the immune system after birth. Dietary supplementation with TGF-β2 at 35 μg/kg/day can increase IgG1 and IgG2a production and decrease the ratio of Th1/Th2, indicating that TGF-β2 is involved in promoting the maturation of immune development [98]. Breast milk TGF-β2 levels were also negatively correlated with plasma viral load levels [99]. These results suggested that TGF-β2 plays an important role in the regulation of immune cell responses and the maintenance of immune system homeostasis.

### 3.5. Tumorigenesis

TGF-β2 has been implicated in the development of a variety of cancers, including lung cancer, gastrointestinal stromal cancer, pancreatic cancer, breast cancer, glioma, melanoma, and so on [107,108,109]. Acidosis is a common tumour microenvironment that enables malignant cell proliferation and metastasis [110]. Cancer cells adapted to acidosis could accumulate lipid droplets in the cytosol, which support cancer cell survival and invasiveness. The inhibition of TGF-β2 with antisense oligonucleotide and TβR1 inhibitor prevented lipid droplet formation and significantly reduced cancer cell invasion. These effects were reversed with the replenishment of recombinant TGF-β2. The researchers concluded that TGF-β2 is a key driver of the acidosis-induced lipid metabolism rewiring that is important for cancer cell progression [100]. Culturing mesothelioma cells in an acidic medium increases TGF-β2 secretion, which in turn leads to lipid droplet accumulation in dendritic cells, altering metabolic reprogramming, migratory capacity, the T cell response, and efficacy of anticancer vaccination, which was reversed with SB431542 (TGF-β receptor inhibitor) [101]. The metastasis of lung cancer cells under acidification is associated with miR-7-5p, which is decreased in transgenic mouse models with lung cancer. It can target the 3′UTR of *Tgfb2* mRNA. Human lung tumour samples also show reduced expression of miR-7-5p and increased levels of TGF-β2. The researchers identified that acidic pH-enhanced lung cancer metastasis was regulated by the miR-7/TGF-β2 axis [53]. The metalloenzyme tartrate-resistant acid phosphatase (TRAP/ACP5) has recently gained prominence due to its association with clinically relevant cancer parameters. Interestingly, neutralising antibody-mediated TGF-β2 blockade could limit TRAP-dependent proliferation and migration in breast cancer cells [102].

In addition to the tumour-promoting effects, the anti-tumour activities of TGF-β2 have also been reported. For example, miR-193a promotes the proliferation and metastasis of pancreatic cancer cells by inhibiting the TGF-β2/TβR3 pathway. In contrast, the restoration of TGF-β2/TβR3 signalling or knockdown of miR-193a suppressed the repopulation and metastasis of pancreatic cancer cells [103]. GPC3 was found to be elevated in the serum of hepatocellular carcinoma patients, which plays a crucial role in inducing cell proliferation. GPC3 knockdown-mediated cell cycle inhibition could be recapitulated with the addition of human recombinant TGF-β2, suggesting the possible involvement of TGF-β2 in the growth inhibition of hepatocellular carcinoma cells [67]. From these studies, we can conclude that TGF-β2 indeed plays a crucial role in tumour cell proliferation, metastasis, and invasion; however, in vivo practice requires careful evaluation based on different cell types and pathological environments.

## 4. Conclusions and Remarks

In this review, we summarised the expression, protein processing and secretion, expressional regulation, signalling pathways, and biological roles of TGF-β2 in physiology and pathology. Over the past years, TGF-β2 has literally been considered an analogue of TGF-β1. However, recent studies have gained more insights into the expression, activation, and biological functions of TGF-β2 that are significantly different from TGF-β1. For example, Compared with TGF-β1, the expression of TGF-β2 is more prone to be altered with greater magnitude when exposed to exogenous stimuli. TGF-β2 has high binding affinity to TβR3 and TβR2-B, while TGF-β1 is more inclined to interact with TβR2. The tissue distribution and cell typic expression of TGF-β1and TGF-β2, as well as corresponsive receptors, are also different, which may determine the phenotypic discrepancy between *Tgfb1^−/−^* and *Tgfb2^−/−^* mice. However, much remains to be understood about the precise mechanism of its role in specific diseases. For example, although *Tgfb2^−/−^* mice show remarkable defects in heart and vessel development, the role of TGF-β2 in adulthood cardiovascular diseases is less investigated. Basic studies about the role of TGF-β2 in the urogenital system, palate, and inner ear are also sparse. Studies based on tissue-specific *Tgfb2^−^*^/*−*^ animals are required for a better understanding of the functions of TGF-β2 under physiological and pathological conditions.

## Figures and Tables

**Figure 1 cells-12-02739-f001:**
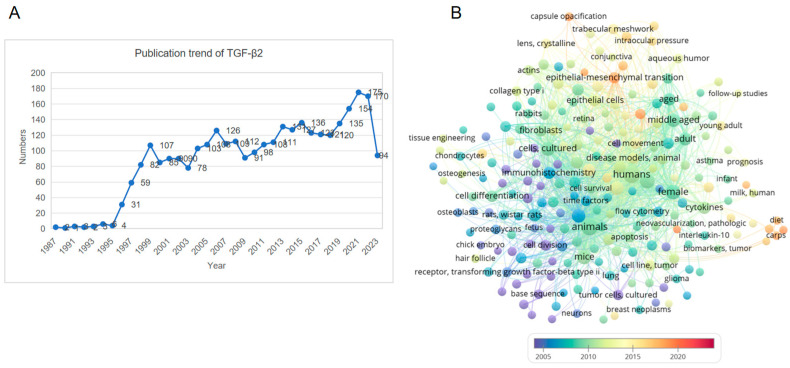
Statistics of TGF-β2-related publications and network visualisation of keywords. (**A**) The trend in TGF-β2 publications from the time of discovery to the present day. (**B**) The result of keyword analysis of TGF-β2-related literature using VOSviewer bibliometrics software, https://www.vosviewer.com/ (accessed on 29 November 2023). The color of dots and lines represent the average year of publication for the corresponding term.

**Figure 2 cells-12-02739-f002:**
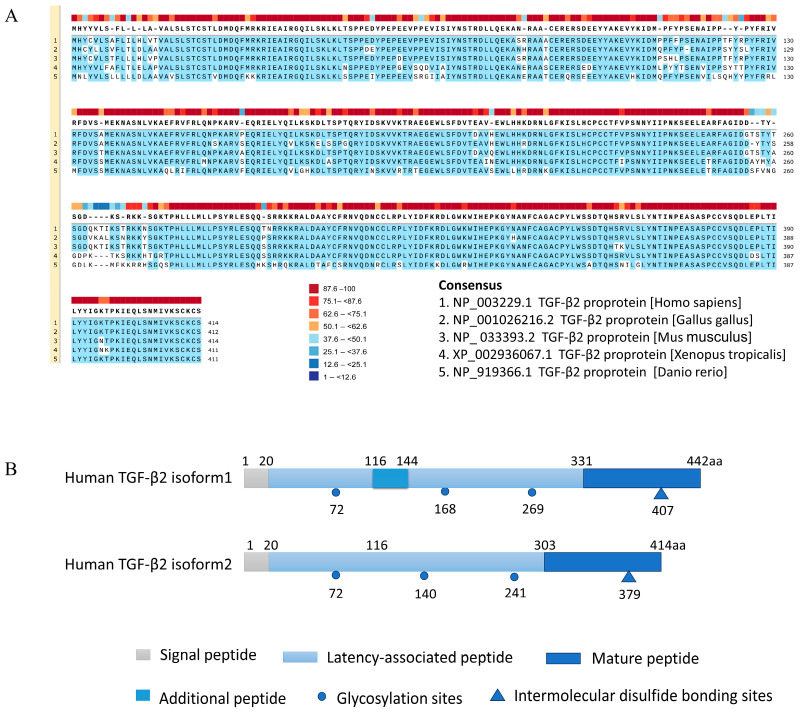
The protein structure and sequence alignment of TGF-β2 between species. (**A**) Conservative analysis of TGF-β2 in Homo sapiens, Gallus gallus, Mus musculus, Xenopus tropicalis, and Danio rerio using SnapGene with a threshold of 90%, highlighted in blue. Sequence conservation was shown by colored blocks on the top. (**B**) Schematic diagram of the structure of the amino acid sequence of the two precursor proteins of TGF-β2 in Homo sapiens.

**Figure 3 cells-12-02739-f003:**
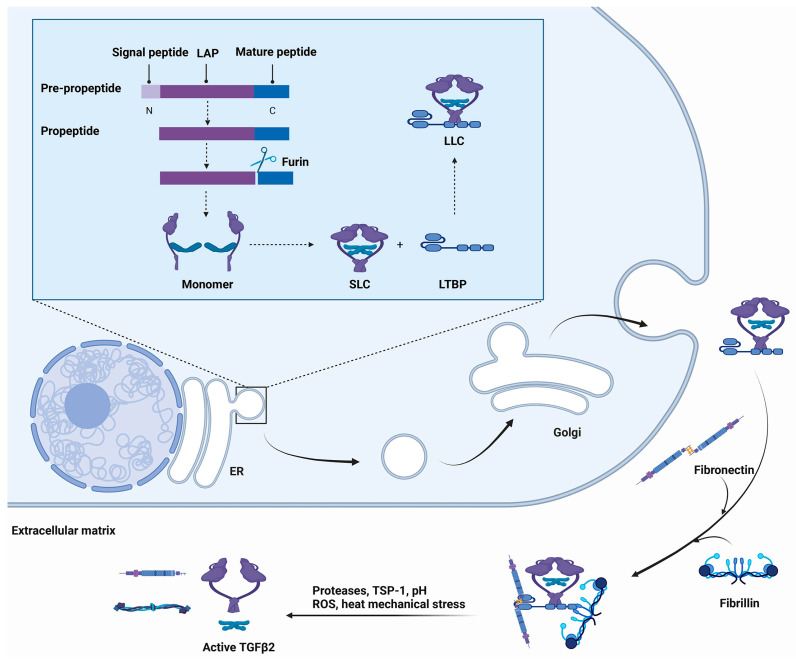
A schematic model of the processing and secretion of TGF-β. The sequential biochemical events are shown from the top left to bottom left, indicated with black arrows. In the lumen of the endoplasmic reticulum (ER), the signal peptide of TGF-β is cleaved by signal peptidase to produce a propeptide, which is further cleaved by furin protease. The latent-associated peptide (LAP) and the mature peptide form a dimerised complex called the small latent complex (SLC). The large latent complex (LLC) is then formed by cross-linking SLC with LTBP with disulfide bonds. LLC is transported to the Golgi apparatus and then released into the extracellular matrix (ECM) where LLC is cross-linked to fibrillin and fibronectin. Upon stimulation by activation-related factors, the complex can be depolymerised to release the active TGF-β2 ligand.

**Figure 4 cells-12-02739-f004:**
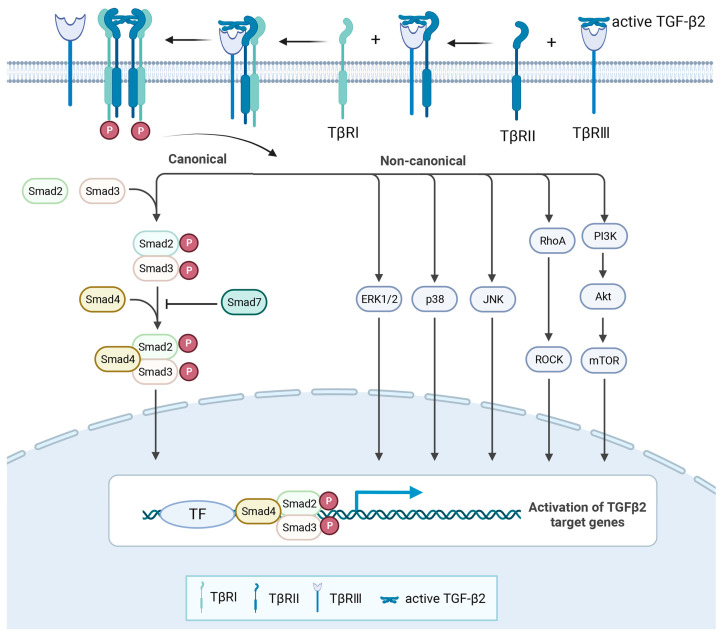
The signal transduction of TGF-β2. In cells with TβR3 receptors, active TGF-β2 firstly binds to TβR3 and then recruits TβR2 and TβR1. Finally, TβR3 is dissociated from the above complex and eventually forms the TGF-β2/TβR1/TβR2 complex. Intracellular signalling is mediated by both canonical and non-canonical signalling pathways that regulate the expression of target genes.

**Table 1 cells-12-02739-t001:** A comparison of the phenotypes of TGF-β1, -β2, and -β3 knockout mice [6,7,8,9].

Organ	*Tgfb1* * ^−^ * * ^/^ * * ^−^ *	*Tgfb2* * ^−^ * * ^/^ * * ^−^ *	*Tgfb3* * ^−^ * * ^/^ * * ^−^ *
Heart	Generalised and extensive infiltration of inflammatory cells involving the pericardium, the myocardium, and the endocardium of the atria and ventricles	Congenital structural defects of the heart, including the arterial outflow tract, aortic and pulmonary orifices, atrioventricular valves, ventricular septum, and myocardium	NA
Lung	Lymphocytic and plasmacytic infiltration	Conducting airways collapsed	Branching morphogenesis and respiratory epithelial cell differentiation defects
Liver	Granulocyte and lymphocyte infiltration, multifocal hepatic necrosis and microgranulomas	NA	NA
Pancreas	Lymphocytic and plasmacytic infiltration	NA	NA
Stomach	Neutrophil and eosinophil cell infiltration	NA	NA
Salivary gland	Lymphocytic and plasmacytic infiltration	NA	NA
Striated muscle	Lymphocytic and plasmacytic infiltration	NA	NA
Brain	Granulocyte and lymphocyte infiltration	NA	NA
Eye	Conjunctivitis, ocular striated muscle inflammation, lacrimal gland inflammation	Hypercellular infusion in the posterior chamber, hyperplastic retina	NA
Skeletal	NA	Limb laxity, spina bifida occulta, sternum malformations, abnormal curvature of the ribs	NA
Craniofacial	NA	Retrognathia, dysmorphic calvaria	NA
Urogenital	NA	Agenesis renal pelvis, testicular ectopia and hypoplasia, ectopia of the uterine horns, degeneration of kidney tubular epithelium, adrenal ectopia	NA
Palate	NA	Cleft palate (partial penetrance, extensive palate cleft)	Cleft palate (full penetrance, soft palate cleft not involved)
Inner ear	NA	Absent spiral limbus and Rosenthal’s canal, undifferentiated inter-dental cells, partially canalised scala vestibuli	NA
Hair follicle	Slightly advanced hair follicle formation	Profound delay in hair follicle morphogenesis	NA
Other phenotypes	Slight enlargement of lymph nodes, smaller spleen and less distinct white pulp	Congenital cyanosis	NA

NA, not available.

**Table 4 cells-12-02739-t004:** The physiological and pathological effects of TGF-β2 or TGF-β2 inhibition.

Organs	Consequences	Treatments	Reference
Eye system	Induces EMT (EMT of retinal pigment epithelial cells is a key mechanism in proliferative retinal diseases)	Human retinal pigment epithelium cells (20 ng/mL TGF-β2)	[79,80]
	Induces senescence and EMT (Degenerative changes in the retinal pigment epithelium play a critical role in the progression of age-related macular degeneration)	Human retinal pigment epithelial cell and human lens epithelial cells (10 ng/mL or 12.5 ng/mL TGF-β2)	[81,82]
	Promote autophagy and EMT (Autophagy plays an important role in fibrotic cataracts)	Rabbit lens epithelial cells and human lens epithelial cells (5 ng/mL or 10 ng/mL TGF-β2)	[83,84]
Cardiovascular system	Promote cardiac fibrosis (Loxl2 stimulates cardiac fibrosis by inducing TGF-β2; MiR-29b-3p and miR-29c-3p inhibit cardiac fibrosis by targeting Tgfb2)	Loxl2-treated TAC hearts and mouse cardiac fibroblasts from Ang-II-infused Mif-KO mice	[56,85]
	Be associated with aortic aneurysm formation	TGF-β2 is elevated at higher levels in the conditioned medium from fibulin-4 deficient mice aortic smooth muscle cells, aortic lysates, and blood	[86]
	Reduce inflammation and matrix degradation and may be involved in maintaining plaque stability	RAW 264.7 cells and human THP-1 blood monocytes (5 ng/mL TGF-β2)	[87]
Motor system	Increase chondrocyte communication, early chondrogenesis, maintain phenotype, and inhibit differentiation	Mouse chondrocytes and human osteoarthritic cartilage (5/10/25 ng/mL TGF-β2)	[60,88,89,90]
	Inhibit skeletal muscle satellite cell differentiation and proliferation and promote apoptosis	Chicken skeletal muscle satellite cells (Inhibition of TGF-β2 by miR-200a-3p)	[91]
	Downregulate bone formation	Human dental follicle stem cells (1 μg/mL TGF-β2 inhibitor)	[92]
	Upregulate gene expression of collagens, extracellular matrix molecules, and growth factors associated with tendon healing	Mouse tendon-derived stem cells and equine bone marrow-derived mesenchymal stem cells (1 ng/mL TGF-β2)	[93,94]
Immune system	Affect immune cell proliferation and apoptosis	MCF-7 cells, RAW 264.7 cells, lamprey supraneural myeloid body cells, and peripheral blood leukocytes (0.01/0.1/1/10 ng/mL TGF-β2)	[95]
	Maintain brain immune homeostasis by regulating the chemokine receptor-modulated immune response in microglia	Mouse NG2 glial cells (5 ng/mL or 10 ng/mL TGF-β2)	[96]
	Redirect inflammatory monocyte-derived cells in central nervous system autoimmunity	Neural precursor cells from *Tgfb2tm1Doe* mice (*Tgfb2^−/−^*) and bone marrow-derived dendritic cells (0.01/1/10/100 ng/mL TGF-β2)	[97]
	Promote the maturation of immune development	G15 pregnant Wistar rats (TGF-β2 at 35 μg/kg/day)	[98]
	Have an impact on antiviral immunity	HIV breast feeding women	[99]
Tumor	Sustain acidic tumor microenvironment- induced lung cancer metastasis	Human lung cancer cells (5 μg/mL of anti-TGF-β2 antibody)	[53]
	Inhibit growth of hepatocellular carcinoma cells	Human hepatocellular carcinoma cells (1 ng/mL or 5 ng/mL TGF-β2)	[67]
	Support acidosis-driven EMT and the metastatic spreading of cancer cells	Human cervix SiHa, pharynx FaDu, colorectal HCT-116, and HT-29 cancer cell lines (10 µM TGFβ2-specific antisense oligonucleotide)	[100]
	Decrease dendritic cells migratory potential and activation and the anticancer immune response	Mouse bone marrow-derived dendritic cells (5 µM TGF-β receptor inhibitor)	[101]
	Mediate the effects of TRAP-dependent proliferation and migration in breast cancer cells	MDA-MB-231 breast cancer cell line (0.25 μg/mL TGF-β2 neutralising antibody)	[102]
	Block pancreatic cancer repopulation and metastasis	Human pancreatic cancer cell lines (Inhibition of TGF-β2 by miR-193a)	[103]

NA, not available.

## Data Availability

Not applicable.

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
