# Peer review of "The Research Progress in Transforming Growth Factor-β2"

_cells, 2023, doi:10.3390/cells12232739_

Round 1
Reviewer 1 Report
Comments and Suggestions for Authors
The authors contributed a well-organized and comprehensive review on the biological mechanisms of pathological relevances of TGF-b2 which is a important cytokine playing multi-faceted roles in biological development and function regulations. Although this review is quite informative and insightful, authors can do the following to further increase the impact of this article:
1. The authors mentioned the relations between TGF-b2 and pathogenesis of several diseases but only provided very limited information regarding how to use TGF-b2 as a potential therapeutic targets for treating these diseases.
2. Many of the citations were from before 2010 or even 2000. Could author update the references to include some more recently-published findings of TGF-b?
Reviewer 2 Report
Comments and Suggestions for Authors
The authors Wang et al have compiled a comprehensive review of the recent advances in research of Transforming growth factor-beta 2 (TGF-β2). This manuscript is well structured and laid out and I have only two comments.
- in Table 2 please do not use contractions ("can't activate"), rather use a more appropriate term.
- You mention the activation of TGF beta proteins through high heat ranging from 65°C to 100°C. Protein activation is reached at 80°C which both puzzling, and intriguing. Does this also refer to biological activity and reactivity? Please elaborate a bit more on the biological relevance of this data, if there is information available. I assume it was only cell culture-derived data but is there more info about heat activation of TGF beta 2 protein e.g. in high fever (40°C)?
Reviewer 3 Report
Comments and Suggestions for Authors
The manuscript 'The research progress of transforming growth factor-B2" is a very interesting manuscript aimed to identified role TGfbeta2 in cellular biology and different biological processes.
Several comments:
The authors could reduce the length of manuscript if will remove unnecessary modifiers:
1. Follow the text a lot of: Authors et al found that…, Autor show or reportes….., These studies suggest….
2. Missed References: lines: 35, 58-60, 62, 68, 200-202…continue to all text
3. Line 37: Tgfb1-/- mice; not all mice died around 20 days, actually 60% die in uterus.
4. Missed description of Fig 1A in the text
5. Line 78-98: Text is hard to understand, not everything related to TGFbeta2. Figure could help.
6. Part the activation of TGFbeta2: a lot of data not related to TGFbeta2.
7. Table 3 references 75,76 not correct. Refer 78, upregulation in endothelial cells does not mean cardiovascular diseases. Also, initialization of myocardial fibrosis is TGFbeta1.
8. Cardiovascular system: mix human, mice, heart, cardiomyopathy, fibrosis, vascular system. Similar situation with musculoskeletal system.
9. Immune system and inflammation are not the same. 8-9: Difficulties understand “Organ-system: sections due to the absence of construction of logical story.
Comments on the Quality of English Language
NA
